# *Akkermansia muciniphila* alleviates experimental colitis through FXR-mediated repression of unspliced XBP1

Fan Bu,[1,2,3,4] Kaiqing Zhang,[1,2,5,6] Bingbing Song,[1,2,5,6] Linhai He,[1,2,4,5] Zhihua Lu,[1,2,4,5] Xiaomin Yuan,[1,2,4,5] Chen Chen,[1,2,4,5] Feng Jiang,[1,2,4,5] Yu Tao,[1,2,4,5] Wei Zhang,[1,2,4,5] Dan Zhang,[1,2,4,5] Yugen Chen,[1,2,4,5] Qiong Wang[1,2,5,6]

**ABSTRACT** Endoplasmic reticulum (ER) stress-related mucin depletion could be involved in the pathogenesis of ulcerative colitis (UC). *Akkermansia muciniphila (A. muciniphila)* uses mucin as its sole energy source and shows potential in the treatment of colitis. However, the effects and underlying mechanisms of *A. muciniphila* on colonic epithelial ER stress in colitis are largely unknown. Colitis was induced by adding 2.5% dextran sulfate sodium (DSS) in drinking water. Mice were orally administered *A. muciniphila* ($3*10^7$, $3*10^8$ cfu/day) once daily for 10 days during DSS intervention. Ultra high performance liquid chromatography q-exactive orbitrap high-resolution mass spectrometry (UHPLC-Q-Orbitrap-HRMS)-based metabolomic analyses were performed on feces. 16S rRNA sequencing was used to quantify and characterize the gut microbiota of mice. Metabolomic analysis showed that P-hydroxyphenyl acetic acid (p-HPAA), the metabolite with the highest variable importance in projection (VIP) score that was elevated by *A. muciniphila*, was negatively correlated with acetic acid levels and exhibited a potential inhibitory effect on ER stress. Additionally, *A. muciniphila* supplementation decreases the abundance of *Parasutterella,* a genus implicated in bile acid homeostasis. By restoring the levels of deoxycholic (DCA) and ursodeoxycholic acid (UDCA), *A. muciniphila* administration normalized the bile acid pool size and composition altered by colitis. *A. muciniphila* supplementation protected colon shortening and histological injury in wild-type (WT) mice, but not in farnesoid X receptor-null ($FXR^{-/-}$) mice. Mechanistically, our results demonstrate that *A. muciniphila* alleviates DSS-induced colitis by targeting inositol requiring enzyme 1α(IRE1α) and unspliced XBP1 (XBP1u) within the ER stress pathway, with the regulation of XBP1u being FXR-dependent. Supplementation with *A. muciniphila* at appropriate doses may, thus, offer a promising therapeutic strategy for Ulcerative colitis (UC).

**IMPORTANCE** UC is a chronic inflammatory disease in which inflammation begins in the rectum and extends proximally throughout the colon. *A.muciniphia* is significantly reduced in UC patients and shows promise as a next-generation probiotic. However, the mechanisms behind its protective effects are not fully understood. Our study reveals that *A. muciniphila* alleviates experimental colitis by reshaping the gut microbiome and correcting imbalances in bile acid metabolism. Crucially, we identify a novel mechanism where *A. muciniphila* acts through the host bile acid receptor FXR to suppress a specific ER stress pathway (XBP1u) in colon cells, thereby helping to restore the intestinal barrier. These findings provide a scientific basis for using *A. muciniphila* as a targeted therapeutic strategy for UC.

**KEYWORDS** *Akkermansia muciniphila*, ulcerative colitis (UC), farnesoid X receptor (FXR), endoplasmic reticulum (ER) stress

**Peer Reviewer** Yi Yin, Nanjing Drum Tower Hospital, Nanjing, China

Address correspondence to Qiong Wang, qiongwang@njucm.edu.cn, or Yugen Chen, yugen.chen@njucm.edu.cn.

Fan Bu and Kaiqing Zhang contributed equally to this article. Author order was determined both alphabetically and in order of increasing seniority.

The authors declare no conflict of interest.

See the funding table on p. 18.

UC is a chronic inflammatory disease in which inflammation begins in the rectum and extends proximally throughout the colon, with major symptoms including blood in the stool, diarrhea, and abdominal pain (1, 2). Although medications are the most effective way for UC treatment, only approximately 40% of patients achieve clinical remission, highlighting the need to explore new treatment strategies (3). *A. muciniphila* has been widely regarded as a promising candidate of next-generation probiotics since it promotes the integrity of the mucosal layer, thereby creating a healthy environment for intestinal epithelial cells (4, 5). Ample studies have showed that *A. muciniphila* was significantly reduced in UC patients and UC animal models, and the intervention of *A. muciniphila* could significantly improve DSS-induced colitis in mice (6–9); however, the mechanisms underlying *A. muciniphila* supplementation against UC are largely unknown.

As a mucin-degrading bacterium, *A. muciniphila* feeds on mucin and produces specific byproducts such as acetate and propionate to feed other beneficial gut bacteria to make butyrate, providing energy source for mucus-secreting goblet cells and intestinal epithelial cells. Recent studies have revealed that aberrant mucin assembly in mice causes ER stress and spontaneous inflammation resembling UC (10). ER is the main site of protein synthesis, folding, lipid synthesis, and carbohydrate metabolism (11). Under ER homeostasis, binding immunoglobulin protein (BiP) proteins bind to ER stress sensors (three ER transmembrane proteins, activating transcription factor 6(ATF6)/IRE1α/ protein kinase RNA‐like ER kinase (PERK)) and keep them in an inactive state (12). In disease states, protein misfolding and accumulation of unfolded proteins occur, BiP proteins have a higher affinity for misfolded/unfolded proteins, and the aforementioned ER sensors are released, activating ER stress, which, in turn, induces an unfolded protein response (unfolded protein response [UPR], also known as ER stress response) (13–15). ER stress may contribute to the aggravation of UC by disrupting intestinal barrier function and activating intestinal inflammatory responses (16, 17). These data reveal that the proven efficacy of *A. muciniphila* in colitis may be involved in modulating the mucin secretion process and ER restoring.

Gut microbiota homeostasis could be affected by bile acid signaling both directly and indirectly (18). *A. muciniphila,* one of the most related commensal gut bacteria, could efficiently increase bile acid metabolism (19, 20). FXR, the main regulator of maintaining bile acid homeostasis, has been proven to be related to ER stress (21, 22). FXR activation by betulinic acid (23), obeticholic acid (24), and GW4064 (25) has been shown to improve disease phenotype from ER stress. Based on the findings, we speculate that *A. muciniphila* may modulate ER restoring in experimental colitis through FXR signaling.

Here, we show that *A. muciniphila* supplementation could reshape the gut microbiota, modulate bile acid metabolism, and alleviate the colonic epithelial ER stress in DSS-induced colitis. Mechanistically, the anti-colitis effect of *A. muciniphila* was achieved through FXR-mediated suppression of XBP1u and subsequent reduction of β-catenin degradation. Our data demonstrate a significant effect of *A. muciniphila* on experimental colitis through an FXR-dependent mechanism that maintains ER homeostasis, suggesting a novel therapeutic implication for UC.

## MATERIALS AND METHODS

### Drugs and reagents

Dextran sulfate sodium salt (DSS, 36–50 kDa) was purchased from MP Biomedicals (CA, USA). Mucin (from porcine stomach) and isothiocyanate-conjugated (FITC)-dextran (3–5 kDa) were purchased from Sigma-Aldrich (St. Louis, MO, USA). TRIzol reagent was purchased from Life Technologies Inc. (Grand Island, NY, USA).

### Bacterial strains

*A. muciniphila* strain ATCC was cultured in brain heart infusion (BHI) medium in tubes at 37°C in anaerobic chamber. *A. muciniphila* was collected in log phase and diluted

with sterile phosphate-buffered saline (PBS) to $3 \times 10^7$ and $3 \times 10^8$ colony-forming units/mice for gavage. *A. muciniphila* freshly prepared every day for gavage.

## Animal experiments

The care and use of the animals followed the animal welfare guidelines, and all the experimental protocols were approved by the Institutional Animal Care and Use Committee of Nanjing University of Chinese Medicine. The mice were housed in a specific pathogen-free environment controlled for temperature and light (23 ± 1°C, 12 h light/dark cycle) and humidity (45%–65%). Male 6- to 8-week-old wild-type, Fxr-null mice were on a C57BL/6J background. Fxr-null mice were obtained from BRL Medicine Inc., Shanghai. Wild-type or Fxr-null mice were randomly divided into four groups: control group, DSS group, AKK-L group, and AKK-H group. Mice in the DSS group, AKK-L group, and AKK-H group were treated with 2.5% of DSS in their drinking water for 10 days, respectively. The mice were supplemented daily with 200 µL of PBS (vehicle), AKK-L ($3*10^7$ CFU/mice), or AKK-H ($3*10^8$ CFU/mice) by intragastric gavage for 10 consecutive days. On the 11th day, mice were anesthetized by inhalation of 3%–5% isoflurane (vol/vol) delivered through a precision vaporizer using medical-grade oxygen at a flow rate of 1–2 L/min. Blood samples were collected from mice posterior orbital plexus and centrifuged at $3,000 \times g$ for 10 min, and the serum was obtained and stored at −80°C. Euthanasia was performed via cervical dislocation. The colon was obtained to measure the colon length. One centimeter of distal colon tissue was collected for histologic examination. The remaining intestinal tube was cut longitudinally, the intestinal mucosa was quickly scraped with a glass slide, and the tube was stored at −80°C for RNA extraction. The blood samples were also collected, set at room temperature for 30 min, and then centrifuged at 3,500 rpm for 10 min at 4°C to obtain serum. Cecal luminal samples were harvested from mice for comprehensive profiling of gut microbiota and metabolite quantification. Moreover, this study is reported in accordance with ARRIVE guidelines (https://arriveguidelines.org/).

## Disease activity index（DAI）

The changes in DAI were measured using the following criteria: (i) weight loss (%), (ii) stool consistency, and (iii) blood in faces as previously described (Table 1) (26).

## Analysis of fecal metabolomics using UHPLC-Q-Orbitrap-HRMS

Sample extraction process: remove the sample from the −80°C refrigerator and thaw it on ice; the whole process is performed on ice, weigh the sample 20 mg (±1 mg), add 70% methanol 400 µL, and vortex for 3 min. Sonicate in the ice water bath for 10 min, remove the sample vortex for 1 min, and let stand in the −20°C refrigerator for 30 min. Centrifuge at 12,000 r/min for 10 min at 4°C, take 300 µL of supernatant, centrifuge the supernatant at 4°C, centrifuge at 12,000 r/min for 3 min, and take 200 µL of supernatant for analysis.

Liquid chromatography-tandem mass spectrometry (LC-MS/MS) analyses were performed using a UHPLC system (1290, Agilent Technologies, Hilden, Germany) with a UPLC HSS T3 column (2.1 mm × 100 mm, 1.8 µm) coupled to a Q Exactive benchtop Orbitrap mass spectrometer (Orbitrap MS; Thermo, Waltham, MA, USA).

The mobile phase A comprised of 0.1% formic acid in water for positive ionization mode, and 5 mmol/L ammonium acetate in water for negative ionization mode, and

**TABLE 1** Disease activity index

| Weight loss (%) | Stool consistency | Occult blood | Score |
| --- | --- | --- | --- |
| None | Normal | Negative | 0 |
| 1–5 | | | 1 |
| 5–10 | Loose stools | Haemoccult+ | 2 |
| 10–20 | | | 3 |
| >20 | Diarrhea | Gross bleeding | 4 |

the mobile phase B comprised of acetonitrile. The elution gradient was set as follows: 0 min, 1% B; 1 min, 1% B; 8 min, 99% B; 10 min, 99% B; 10.1 min, 1% B; and 12 min, 1% B. The flow rate was set to 0.5 mL/min, and the injection volume was 2 µL. The Q Exactive mass spectrometer was used because of its ability to acquire MS/MS spectra on an information-dependent acquisition (IDA) mode during the LC-MS/MS experiment. The data acquisition software (Xcalibur 4.0.27, Thermo) continuously evaluated the full scan survey MS data in the IDA-based mode because it collects and triggers acquisition of MS/MS spectra depending on the preselected criteria. Electrospray ionization (ESI) source conditions were set as follows: sheath gas flow rate was 45 Arb; Aux gas flow rate was 15 Arb. Capillary temperature was 320℃, full Ms resolution was 70,000, MS/MS resolution was 17,500, collision energy was 20/40/60 eV in normalized collisional energy model, and spray voltage was 3.8 kV (positive mode) or −3.1 kV (negative mode).

Full MS raw data files including retention time alignment, peak detection, and peak matching were converted to mzML format using ProteoWizard and processed using R package XCMS. Afterward, the data files were filtered based on the following criterion: sample numbers with metabolites that were less than 50% of all sample numbers in a group. Subsequently, normalization to an internal standard for each sample was conducted, and missing values were replaced by half of the minimum value observed in the data set by default. The preprocessing results generated a data matrix comprising retention times (RTs), mass-to-charge ratio ($m$/z) values, and peak intensity.

In the metabolomic analysis, the VIP score is derived from Orthogonal Projections to Latent Structures-Discriminant Analysis(OPLS-DA) and reflects the contribution of each metabolite to the separation between groups. A VIP > 1.0 is generally considered statistically significant. Fold Change(FC) indicates the ratio of the mean intensity of a metabolite between two groups, with |FC| > 1.5 often used as a threshold for biological relevance.

## 16S rDNA gene high-throughput Sequencing

The V3-V4 variable region of the bacterial 16S rRNA gene was amplified by F338 (5′-ACT CCTACGGGAGGCAGCA-3′) and R806 (5′-GGACTACHVGGG TWTCTAAT-3′). On the Illumina MiSeq platform, the extracted PCR products were analyzed by isomolecular 250-bp double-terminal sequencing. The original pyrophosphate sequence was uploaded to the NCBI Data Center database SRA (Sequence Read Archive). High-quality sequence merge overlaps generated fastq files. QIIME (version 1.9.1, https://qiime.org/) software was used to multichannel decode and quality control filter the fastq file output. All sequencing and bioinformatics analysis were performed using the Omicsmart online platform (http://www.omicsmart.com).

## Bile acid analysis

All the bile acids (BAs) standards were synthesized by Metabo-Profile lab or obtained from Steraloids Inc. (Newport, RI, USA) and TRC Chemicals (Toronto, ON, Canada). All the BAs and Isotopic internal standards were accurately weighed and prepared in methanol solution to obtain individual stock solution at a concentration of 5.0 mM. Appropriate amount of each BAs stock solution was mixed and stepwise diluted in bile-acid-free matrix (BAFM, serum, or uric) at the concentration of 2,500, 500, 250, 50, 10, 2.5, and 1.0 nM to prepare standard solutions. Additionally, appropriate amount of each BAs stock solution was mixed and stepwise diluted in BAFM at the concentration of 1,500, 150, and 5 nM (high, medium, and low) to prepare quality control samples. Internal standard was added into all standard solutions and quality control samples to monitor data quality and to adjust for the matrix effect. All standard solutions and quality control samples were prepared to have the same concentration Internal standard within [with glycocholic acid (GCA)-d4, taurocholic acid (TCA)-d4, taurochenodeoxycholic acid (TCDCA)-d9, UDCA-d4, cholic acid (CA)-d4, glycochenodeoxycholic acid (GCDCA)-d4, glycodeoxycholic acid (GDCA)-d4, DCA-d4, lithocholic acid (LCA)-d4, and β-CA-d5 all at the concentration of 150 nM].

## Haematoxylin and eosin staining

Distal colon specimens were fixed for 48 h in 4% formalin after mice were sacrificed. Then, the distal colon specimens were paraffin-embedded. Finally, the sections were segmented and stained with H&E, and pathological changes were observed with a light microscope.

## Periodic acid-Schiff and Alcian blue staining

After deparaffinization and rehydration, the sections were stained with PAS/AB. The goblet cells are blue. The number of goblet cells was counted using Image J software and expressed as positive cells per villus.

## Immunohistochemical staining

First, paraffin sections were dewaxed in water; antigen repair was performed, and endogenous peroxidase was blocked. The sections were blocked in serum (same source as secondary antibody), which was followed by primary antibody application, secondary antibody application, DAB color development, nuclear staining, dehydration, and sealing. Finally, the positive expression of mucin-2 (MUC2) and beta-catenin in colonic mucosal epithelial cells was observed under a microscope (27).

## Intestinal permeability

Intestinal permeability was determined by the FITC-dextran assay. Mice were fasted overnight and were administered FITC-dextran (2.5 mg/mice) by enema 4 h before blood collection. The concentration of FITC was determined by spectrophoto fluorometry (490/525 nm).

## Assessment of cytokine level in serum

The level of interleukin-1β (IL-1β) and interleukin-6 (IL-6) were measured using commercial enzyme-linked immunosorbent assay (ELISA) kits according to the manufacturer's instructions.

## Immunofluorescence staining

Immunostaining was performed with the standard protocol using antibodies targeting CHOP, XBP1, IRE1, p-IRE1, beta-catenin. The samples were incubated overnight with the primary antibodies at 4°C, rinsed with PBS, and then probed with the Cy3-conjugated secondary antibody (1:1,000, Cat #ab6939; Abcam) for 1 h at 37°C in the dark. After counterstaining with 4',6-diamidino-2-phenylindole (DAPI), the samples were observed using laser scanning confocal microscopy (Leica, Wetzlar, Germany).

## Quantitative real-time polymerase chain reaction

Total RNA was extracted from colon tissues using TRIzol reagent, and the concentration of RNA was measured and then reverse transcribed according to the manufacturer's instructions using a HiScript 1st Strand cDNA Synthesis Kit (Vazyme, Nanjing, China). cDNA was used for qPCR using SYBR Green Master Mix (Service, Wuhan, China) on an ABI 7500 Fast Real-Time PCR System (Applied Biosystems). Relative amounts of mRNA were calculated using the $2^{-\Delta\Delta CT}$ method, and GAPDH served as the housekeeping gene. The primer sequences are shown in Table 2.

## WB analysis

Total protein was extracted from colon tissues using radioimmunoprecipitation assay (RIPA) lysis buffer and quantitated using a bicinchoninic acid (BCA) protein assay kit. The concentration of extracted protein was quantified by BCA protein assay kit (Beyotime). An equal amount of protein sample was loaded on sodium dodecyl

**TABLE 2** Primers used in the real-time PCR assays

| Gene | Direction | Primer sequences (5′ → 3′) |
|---|---|---|
| IL-6 | Forward | TAGTCCTTCCTACCCCAATTTCC |
| | Reverse | TTGGTCCTTAGCCACTCCTTCC |
| GAPDH | Forward | GTGGAAGGTCGGTGTGAACG |
| | Reverse | CTCGCTCCTGGAAGATGGTG |
| IL-1β | Forward | ACCTTCCAGGATGAGGACATGCA |
| | Reverse | CTAATGGGAACGTCACACACCAG |
| CHOP | Forward | GAATGAAGAGGCGCTGAGG |
| | Reverse | CTCCTGCTCCTTCTCCTTCG |
| ATF6 | Forward | GGAGCCAGGCTGAAAGAGTAG |
| | Reverse | GTGAAGACCCCTAAGCCAGA |
| XBP1 | Forward | GGTCTGCTGAGTCCGCAGAG |
| | Reverse | GAAAGGGAGGCTGGTAAGGAAC |

sulfate-polyacrylamide gel electrophoresis gel and then transferred to a polyvinylidene difluoride membrane (Millipore, Billerica, MA, USA). After the membrane was blocked with skimmed milk, it was incubated with primary antibodies against

XBP1(proteintech, 24168-1-AP, 1:1,000)[XBP1u 32KD and spliced XBP1(XBP1s) 60KD], p-IRE1α(abcam, ab48187, 1:1,000), ATF6(proteintech,24169-1-AP, 1:1,000) ,and β-actin (proteintech, 66009-1-Ig, 1:1,000). The next day, secondary antibodies conjugated with Horseradish peroxidase were probed for 2–3 h. Protein signals were detected with enhanced enhanced chemiluminescence (ECL) reagent based on the manufacturer's instructions.

## Statistical analysis

Graphing was performed using GraphPad Prism (version 9.0, https://www.graph-pad.com). One-way ANOVA analysis of variance was applied to compare differences between multiple groups. When only two groups were compared, Student's $t$-test was conducted. Non-parametric data were analyzed by the Mann-Whitney U test. A value of $P < 0.05$ indicated that the difference was statistically significant. All plots are shown as the mean ± standard error of the mean (SEM). $P<0.05$ was considered statistically significant.

## RESULTS

### *A. muciniphila* altered fecal metabolic composition in DSS-induced mice

Oral gavage with *A. muciniphila* in DSS-induced mice was conducted since the abundance of *A. muciniphila* in UC patients was lower significantly than healthy control (28). Fecal samples from the control, DSS, and *A. muciniphila* (AKK) groups were collected and analyzed by UHPLC-Q-Orbitrap-HRMS in the positive ion mode, which represented physiological status, pathological conditions, and intervening effects. The supervised OPLS-DA analysis was performed among the three groups. As is depicted in Fig. 1A and B, there were significant separations in the metabolic profiles among the three groups.

The volcano plots generated to identify the differentially expressed metabolites were shown in Fig. 1C and D, and there were 125 differentially expressed metabolites identified. Moreover, red dots represented the upregulated metabolites (FC > 1), and green dots represented the downregulated metabolites (FC < −1). FC indicates the ratio of the mean intensity of a metabolite between two groups, with |FC| > 1.5 often used as a threshold for biological relevance. Venn diagrams indicated different and shared metabolites among the three groups. Based on the KEGG pathway enrichment analyses, differentially expressed metabolites between DSS and AKK groups were predominantly enriched in mineral absorption, bile secretion, and protein digestion and absorption (Fig. 1E). In the mineral absorption pathway, we found that hydrogen phosphate was

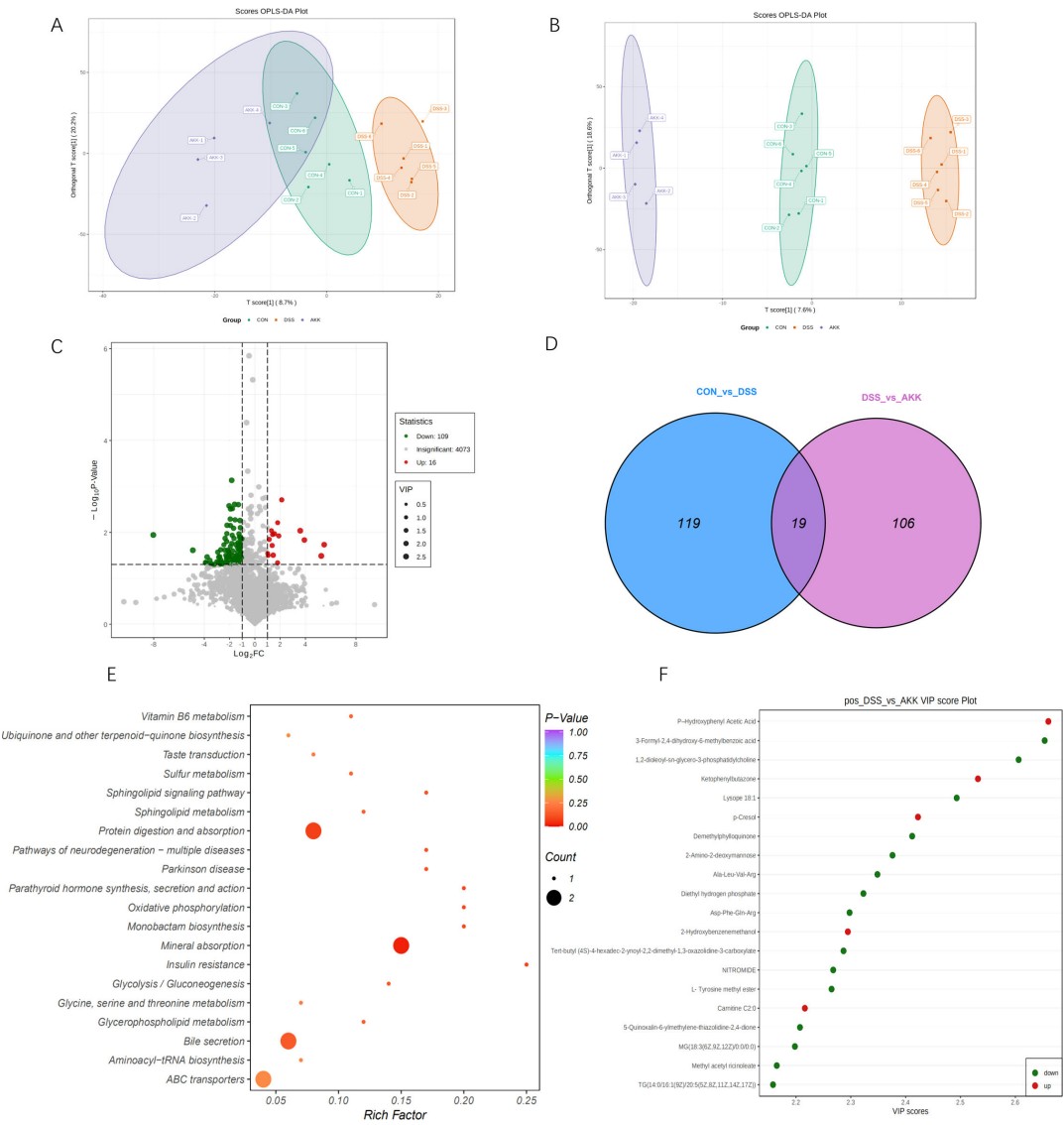

**FIG 1** *A. muciniphila* altered fecal metabolic composition in DSS-induced mice. (A and B) Score plot of OPLS-DA analysis comparing the metabolome profile among three groups. (A) ESI+ mode and (B) ESI− mode. (C)Volcanic plots of fecal metabolic profiling. (D)VENN diagram. (E) Top 20 of the KEGG pathway enrichment analysis of differentially expressed metabolites between DSS and AKK groups. (F) Differentially expressed metabolites between DSS and AKK groups.

significantly downregulated by *A. muciniphila* (*P*=0.026), and hydrogen phosphate had been reported significantly associated with ER stress (29, 30).

The metabolites with features of VIP > 1 and *P* <0.05 were considered potential significant differentially expressed metabolites. The top 20 differentially expressed metabolites in fecal samples between the DSS and AKK groups were shown in Fig. 1F. The intervention of *A. muciniphila* significantly increased the expression of p-HPAA, Ketophenylbutazone, p-Cresol, and Carnitine C2:0 and decreased the expression of 3-Formyl-2,4-dihydroxy-6-methylbenzoic acid, 1,2-dioleoyl-sn-glycero-3-phosphatidyl-choline, Lysope 18:1, Demethylphylloquinone, 2-Amino-2-deoxymannose. Among them, p-HPAA (31) and 1,2-dioleoyl-sn-glycero-3-phosphatidylcholine with the top 3 VIP scores were related to the inhibition of ER stress (32). The analyses result of fecal metabolomics revealed that the activation of ER stress induced by DSS was attenuated by *A. muciniphila*. Moreover, *A. muciniphila* might also have certain impacts on BAs under the experimental colitis setting.

## *A. muciniphila* supplementation modulated the structure of gut microbiota and bile acid dysmetabolism in DSS-induced mice

To quantify and characterize the gut microbiota of mice under different conditions, 16S rRNA sequencing was utilized. The Shannon index was used as a metric to analyze differences in alpha diversity between the groups. ACE and PD-tree index was used as a metric to analyze differences in alpha diversity between the groups (Fig. 2A and B). We observed a decrease to normal level in alpha diversity in the AKK-H group compared to that in the DSS group ($P < 0.05$). Principal coordinates analysis (PCoA) plots were calculated from unweighted_unifrac distances to evaluate the composition of the community, and the results revealed a clear separation among the three groups (Fig. 2C). We also observed a marked separation among the three groups in the system clustering tree (Fig. 2D). The most abundant taxa at the phylum, family, and genus levels are shown in Fig. 2F through H. *A. muciniphila* supplementation significantly increased the relative abundance of *Bacteroidota* and *Desulfovibrio* upon DSS challenge. Notably, *A. muciniphila* supplementation significantly decreased the relative abundance of *Parasutterella* (Fig. 2E), which was found to enhance the catabolism of bile acid (33).

We, therefore, identified a total of 27 fecal BAs across all treated groups using LC/QTRAP-MS. The ratio of secondary to primary BAs was lower in the colitis group compared with those in the controls and was restored to normal levels in the *A. muciniphila*-treated group (Fig. 2I). DSS administration markedly decreased the levels of DCA, chenodeoxycholic acid (CDCA), UDCA, TCDCA, and tauroursodeoxycholic acid (TUDCA) (Fig. 2J). Compared with the colitis group, *A. muciniphila* treatment increased the levels of DCA, UDCA, hyodeoxycholic acid (HDCA), TCA, TCDCA, TUDCA, LCA, and isoLCA. The results demonstrated that *A. muciniphila* treatment regulated the dysbiosis of gut microbiota as well as bile acid dysmetabolism induced by DSS. Based on reported evidence that *Parasutterella* abundance negatively correlates with fecal DCA levels (34) and considering that DCA represents one of the most potent FXR ligands (18), we further investigated the involvement of FXR in the protective effects of *A. muciniphila* against DSS-induced colitis and associated ER stress.

## *A. muciniphila* supplementation prevented DSS-induced colitis in WT mice, while this effect was lost in *Fxr*-null mice

As shown in Fig. 3A through F, the DSS group showed significant weight loss, diarrhea, hematochezia, and other colitis symptoms. Compared to that of the control group, the colon length was markedly shortened in the DSS group, and the colon tissue was characterized by inflammatory cell infiltration, epithelial cell destruction, and mucosal thickening both in WT mice and FXR$^{-/-}$ mice. AKK-H was used to significantly improve weight loss, DAI score, colonic shortening, and restored intestinal epithelial structure induced by DSS in WT mice. Consistent with the symptom observations, *A. muciniphila* treatment did not contribute to weight loss, DAI score, colonic shortening, and the reduction of severe intestinal epithelial damage in FXR$^{-/-}$ mice. These results indicated that AKK-H has an obvious protective effect on DSS-induced colitis in WT mice but not in FXR$^{-/-}$ mice.

To elucidate the inflammatory response in DSS-induced mice, various proinflammatory cytokines were measured in colon tissues at the mRNA level (Fig. 3G through J). The levels of proinflammatory cytokines, including IL-6 and IL-1β, were increased in DSS-induced colitis mice. Quantification of specific cytokines using ELISA showed the same patterns in the regulation of production and secretion of pro-inflammatory cytokines. These elevated proinflammatory cytokines were all decreased by AKK in WT mice not in FXR$^{-/-}$ mice.

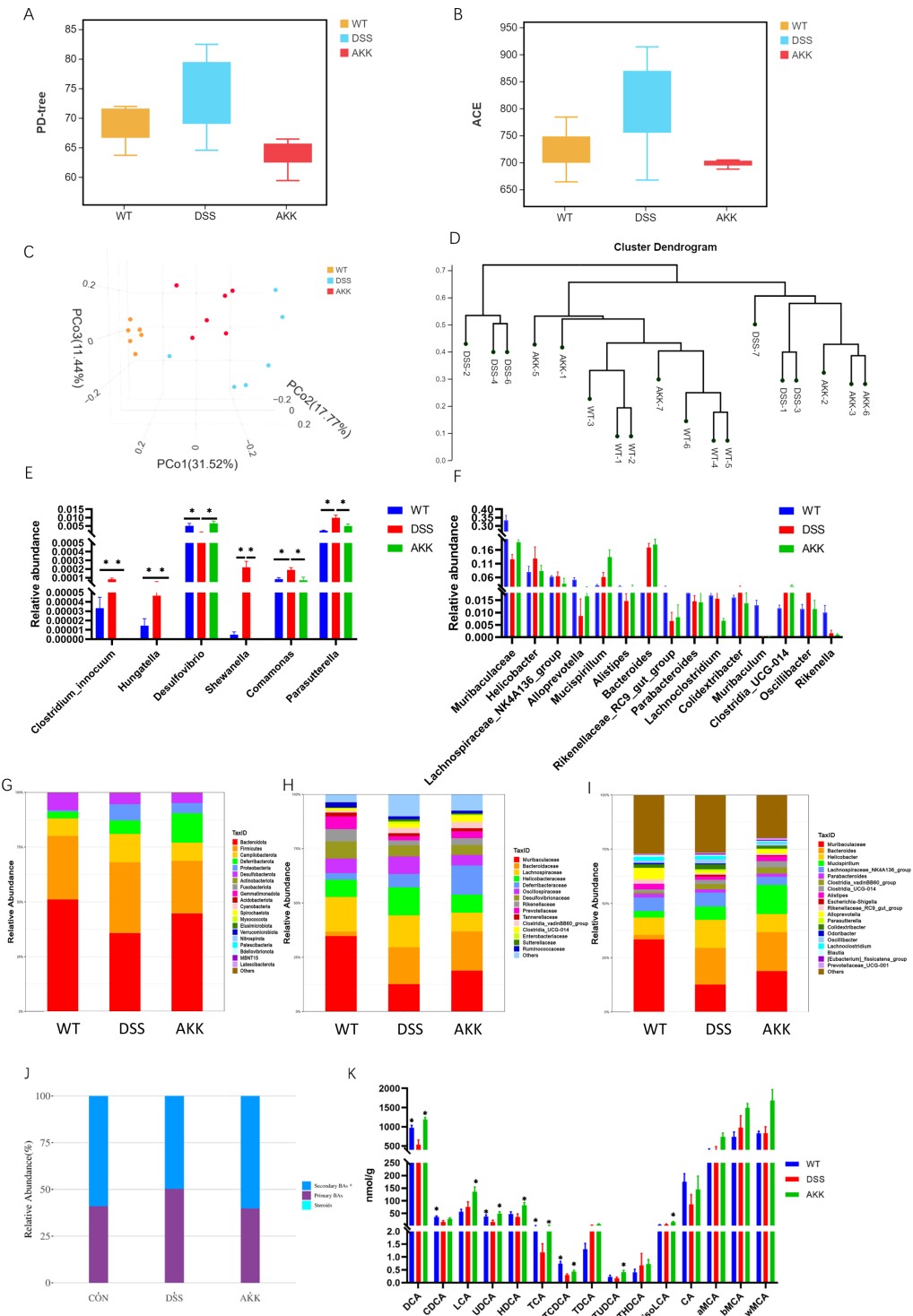

**FIG 2** *A. muciniphila* modulated the structure of gut microbiota and bile acid dysmetabolism in experimental colitis. (A) PD-TREE index. (B) ACE index. (C) pCoA score plot based on unweighted Unifrac metrics. (D) Cluster analysis based on Bray-Curtis metric distances. (E and F) Relative abundance of taxa in different groups. Data are shown as the mean ± SEM. *P < 0.05. (G–I) Relative abundance of taxa at the phylum (G), family (H), and genus (I) levels. (J) Ratios of second/primary BAs in feces. (K) Relative abundance of the significantly altered BAs from different groups. Data are shown as the mean ± SEM. *P < 0.05 vs DSS group.

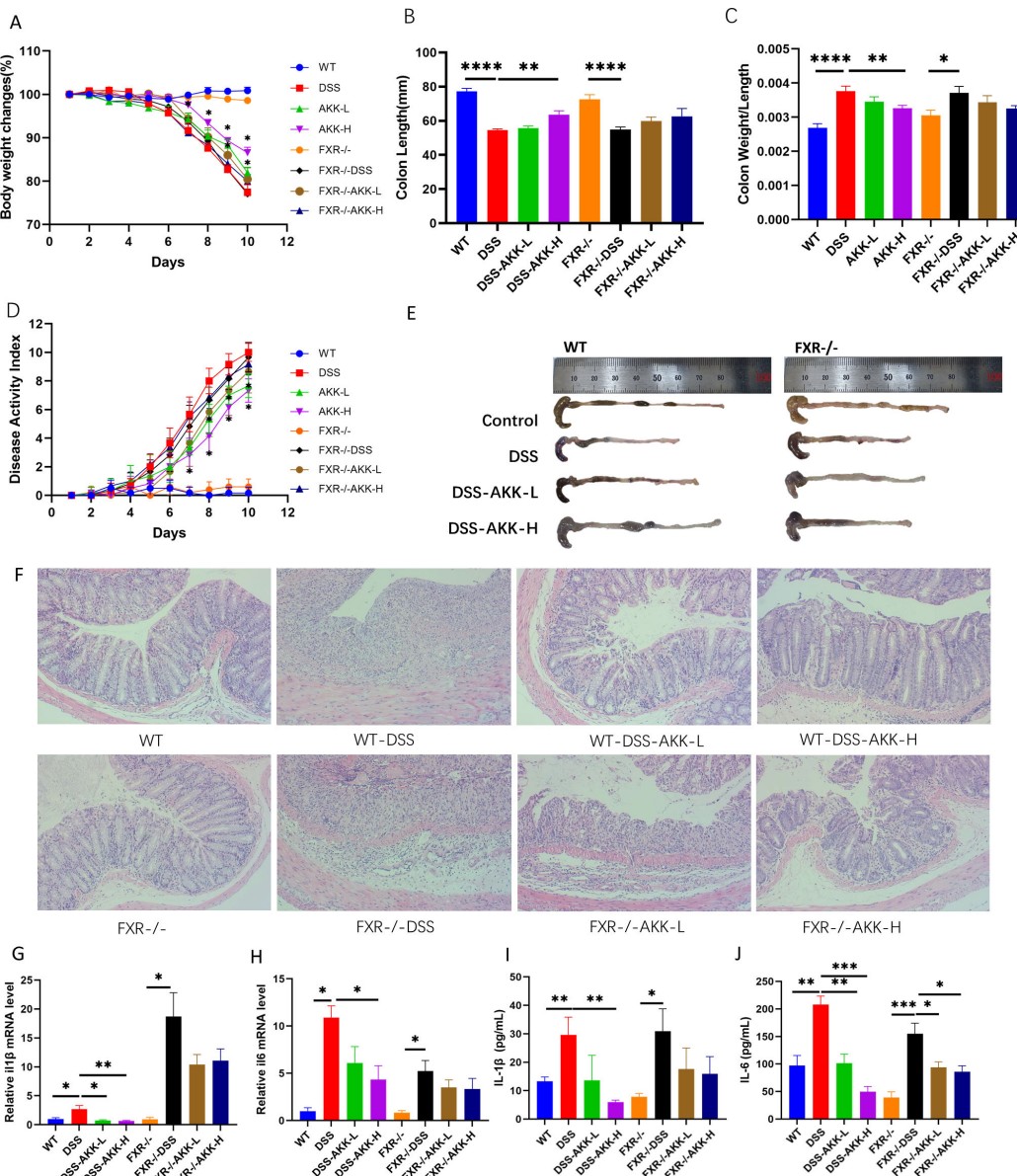

**FIG 3** *A. muciniphila* supplementation ameliorated the symptoms of DSS-induced colitis in mice. (A) DAI score,*P < 0.05. (B) The lengths of colon,**P < 0.01,****P < 0.0001. (C) Colon weight/length,*P < 0.05, **P < 0.01,****P < 0.0001. (D) Weight changes. (E) Macroscopic appearances of colon tissues. (F) Histological changes (H&E staining images of colonic sections at original magnification ×200) (G and H) mRNA quantification of pro-inflammatory cytokines (IL-1β and IL-6). (I and J) Determination of protein productions of the inflammatory cytokines using ELISA (IL-1β and IL-6). The data present the mean ± SEM and *n* = 5/group. *P < 0.05, **P < 0.01, ***P < 0.001.

## *A. muciniphila* supplementation FXR-dependently maintained the intestinal barrier integrity in DSS-induced mice

To understand the effect of *A. muciniphila* on the intestinal barrier integrity of the mice with DSS-induced colitis, intestinal permeability was measured using FITC-dextran as a tracer. The expression of MUC2 in the colon was determined by immunohistochemical staining. Due to mucin produced by goblet cells, we examined the colon in mice from different groups and counted the number of goblet cells by PAS/AB staining. As shown in Fig. 4A through E, serum dextran levels were significantly higher in colitis mice. After treatment of *A. muciniphila*, the levels of FITC-dextran in serum were decreased in WT mice, but not in FXR$^{-/-}$ mice. Immunohistochemical staining results showed that the

content of MUC2-positive cells in DSS-induced mice was significantly lower than that in the control group. PAS/AB staining results revealed the number of goblet cells was significantly decreased in DSS mice compared with that in control mice. *A. muciniphila* supplementation significantly enhanced the expression of MUC2 and increased the number of goblet cells and MUC2-positive cells in each villus. While, in FXR$^{-/-}$ mice, the recovery of the DSS-induced goblet cell injury mediated by *A. muciniphila* was largely suppressed. Meanwhile, the effects of *A. muciniphila* on the expression of MUC2 were also blocked in FXR-deficient mice.

To observe the effect of DSS on ER stress in colon and the role of *A. muciniphila* in regulating ER stress, some markers relevant to ER stress response were examined respectively using qPCR and immunofluorescence assay. As shown in Fig. 5, PCR analysis showed the gene levels of XBP1 and ATF6 were significantly increased after 10 days DSS exposure ($P < 0.05$) in WT mice, and the elevation of XBP1 and ATF6 was significantly decreased by *A. muciniphila* ($P < 0.05$). Meanwhile, the results of immunofluorescence suggested that *A. muciniphila* could decrease the expression of p-IRE1α induced by DSS, indicating *A. muciniphila* alleviated DSS-induced ER stress in colon tissues of mice. Western Blot analysis was conducted to measure the protein expression levels of XBP1 and p-IRE1 in the colonic tissues of mice (Fig. 5). As depicted in Fig. 5, compared to the control group, the phosphorylation level of IRE1 and XBP1u was significantly increased in the DSS group ($P < 0.05$), which was reversed by the treatment of *A. muciniphila*. No statistically significant alterations in ATF6 expression were observed in any of the groups (Fig. S1). Despite a decrease in FXR agonists like DCA after DSS induction, p-IRE1 expression was paradoxically upregulated. This, combined with the observation that DSS induction did not significantly elevate p-IRE1 in FXR-deficient mice, collectively indicates that the suppression of p-IRE1 is FXR-independent. Notably, the activating effects of *A. muciniphila* on the expression of XBP1u were blocked in FXR-deficient mice. These data indicated that the effects of *A.muciniphila* on colitis were involved in the XBP1u ER stress pathway, which was mediated by FXR. While the role of XBP1u in colitis has not been directly established, a prior study (35) demonstrated its function in promoting the ubiquitination-mediated degradation of β-catenin—a key protein for

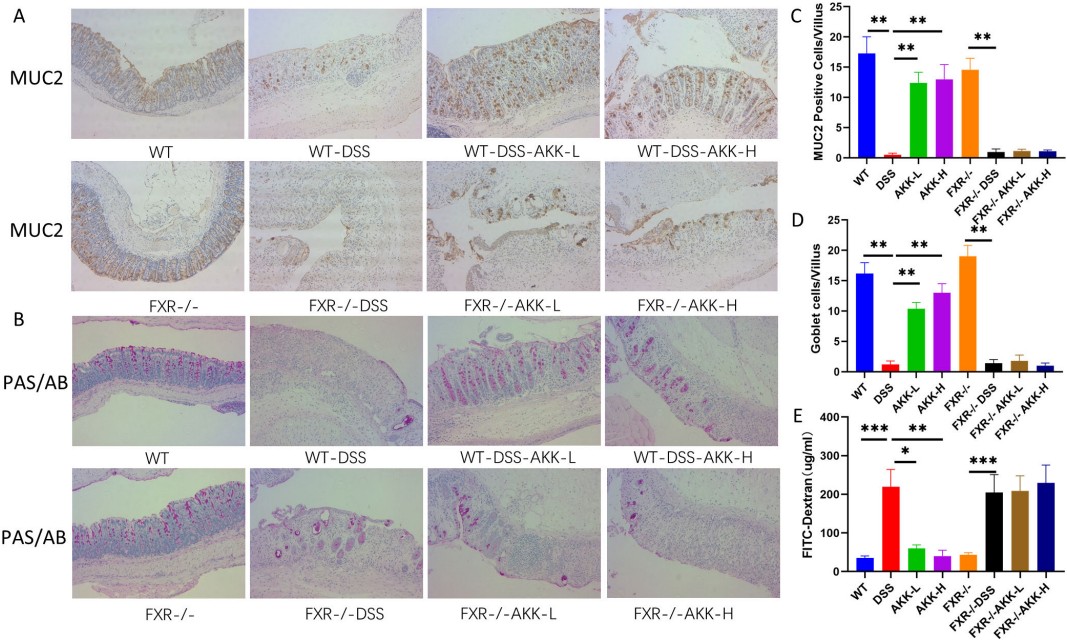

FIG 4   Effects of *A. muciniphila* on the intestinal barrier integrity in DSS-treated mice. (A) Representative images showing the number of colonic MUC2 positive cells (×200 magnification, scale bar 50 µm). (B) Representative pictures showing that colonic specimens stained with PAS/AB (magnification ×200, scale bar: 50 µm). (C) The numbers of MUC2-positive cells in each villus,**$P < 0.01$. (D) Numbers of goblet cells in each villus. **$P < 0.01$. (E) Epithelial permeability of FITC-dextran,*$P < 0.05$, **$P < 0.01$,***$P < 0.001$.

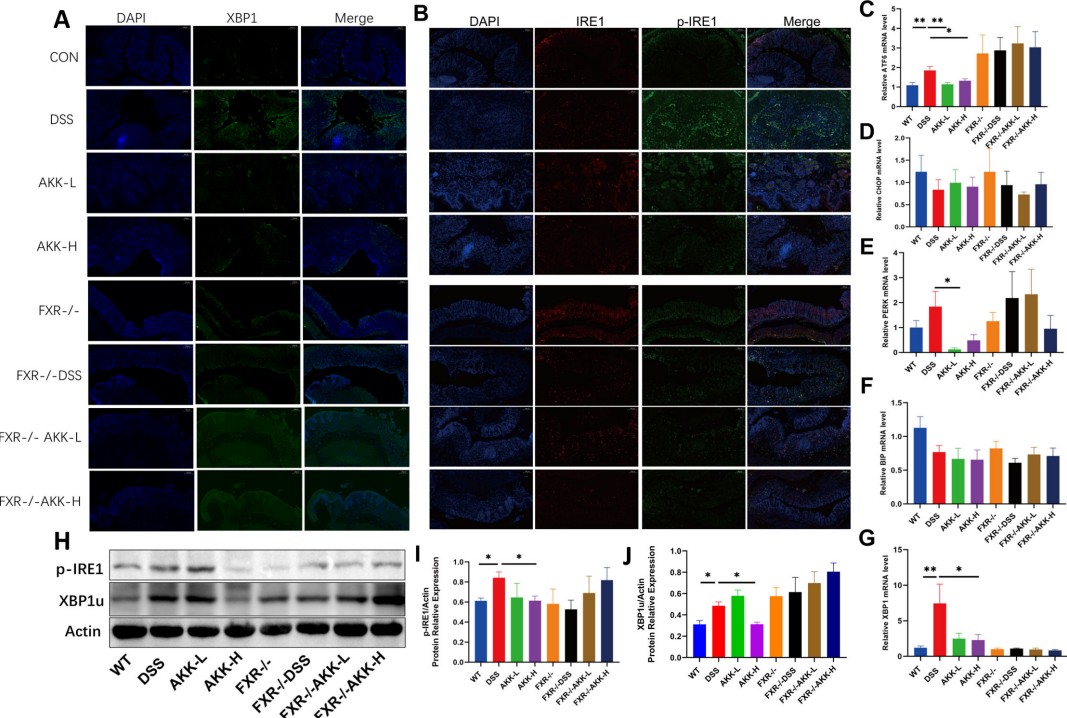

**FIG 5** *A. muciniphila* supplementation FXR-dependently attenuated colonic ER stress in DSS-induced colitis mice. Sections of colonic tissues were immunostained with DAPI and antibodies and then observed under ×200 fluorescence microscope. (A) XBP1 immunofluorescence staining of representative colon tissue. (B) IRE1 and p-IRE1 immunofluorescence staining of representative colon tissue. (C–G) mRNA quantification of ER stress markers (ATF6, CHOP, PERK, Bip, and XBP1). (H) Western blot was used to detect the expression levels of XBP1u and p-IRE1. (I and J) The relative quantitative analysis of XBP1u and p-IRE1 was performed. Data are presented as means ± SEM. *$P < 0.05$, **$P < 0.01$.

maintaining the intestinal barrier under physiological conditions. Given this, we assessed colonic β-catenin levels and observed a significant reduction in DSS-induced colitis mice. This deficit was markedly rescued by A. *muciniphila* supplementation through an FXR-dependent mechanism, as shown in Fig. 6.

## DISCUSSION

In the past few years, the impact of the gut microbiota on UC has received extensive attention. Dysbiosis of gut microbiota or related metabolic dysregulation affects the occurrence and development of UC. Studies from both animal models and clinical trials have shown that *A. muciniphila* is significantly reduced in the UC disease state (6, 7). Our previous study showed that *A. muciniphila* could markedly inhibit the release of intestinal inflammatory cytokines in colitis mice induced by DSS, improve intestinal inflammation, and ameliorate colitis symptoms (9), but the effective dose and potential mechanism of action are not clear.

*A. muciniphila* may promote the metabolism of the mucus layer by increasing the production of MUC2 to maintain the health of intestinal epithelial cells and the integrity of the barrier (27). *A. muciniphila* has also been shown to repair the intestinal barrier *in vitro* (36). Consistent with the previous study, the application of *A. muciniphila* can dose-dependently inhibit inflammation and increase the expression of MUC2 in experimental colitis. KEGG enrichment analysis in mice feces showed that the differentially expressed metabolites were significantly enriched in the mineral absorption pathway before and after *A. muciniphila* treatment, and the level of hydrogen phosphate increased in the DSS group, while *A. muciniphila* treatment could significantly decrease the level of hydrogen phosphate ($P < 0.05$). It has been reported that high levels of hydrogen phosphate promote the occurrence of ER stress (37). Current research on

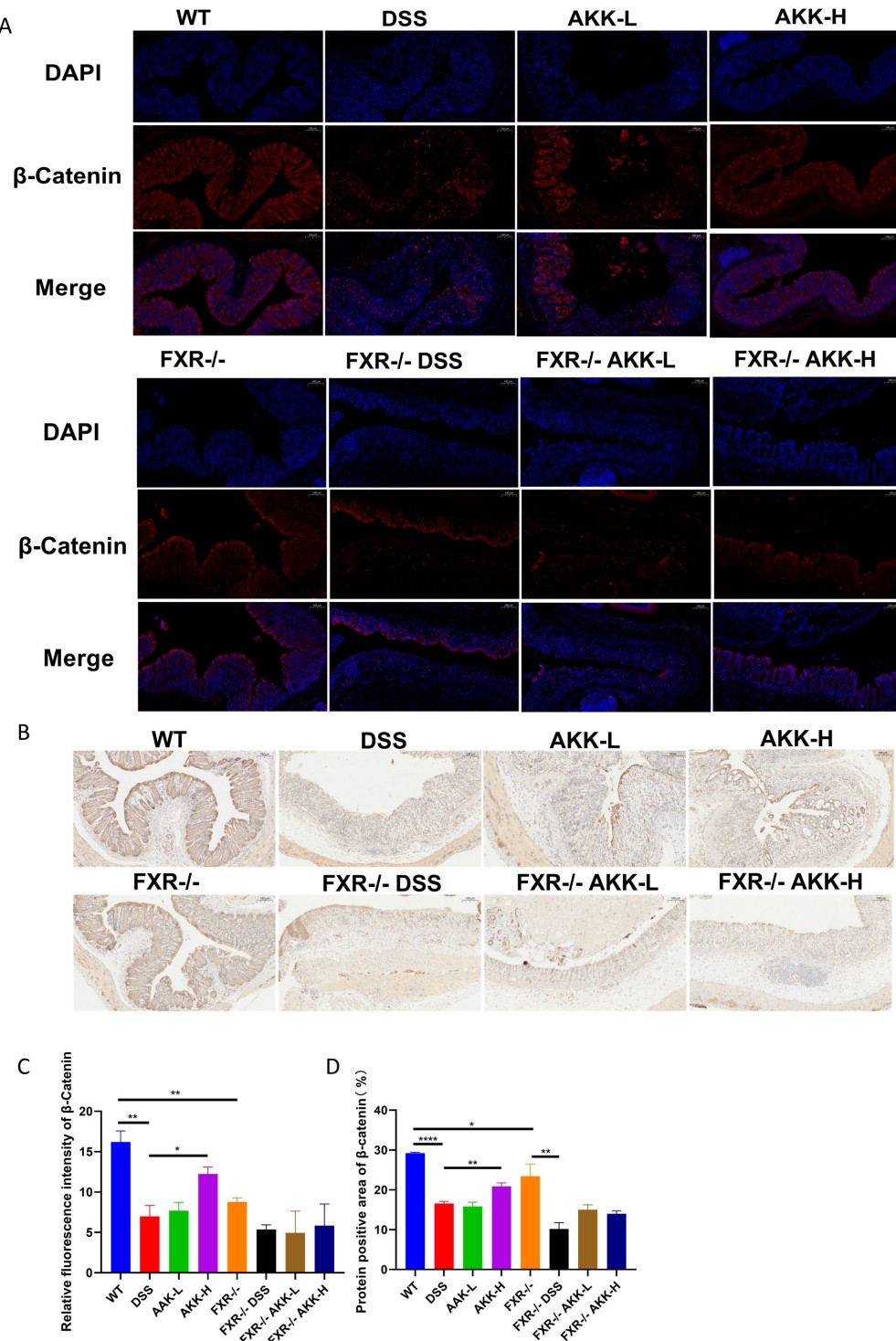

**FIG 6** *A. muciniphila* supplementation FXR-dependently increased the β-catenin levels. (A) β-Catenin immunofluorescence staining of representative colon tissue. (B) The number of colonic β-catenin positive cells in DSS-treated mice. Representative images (×200 magnification). (C) Relative fluorescence intensity of β-catenin,*$P < 0.05$, **$P < 0.01$. (D) Protein positive area of β-catenin. Data are presented as means ± SEM, *$P < 0.05$, **$P < 0.01$,****$P < 0.0001$.

the direct inhibition of ER stress by p-HPAA remains in its preliminary stages. Studies suggest that p-HPAA may indirectly improve intestinal homeostasis by activating the aryl hydrocarbon receptor (AhR) pathway. p-HPAA is a product of gut microbial

metabolism of dietary aromatic amino acids (such as tyrosine) and belongs to the family of AhR ligands. The activation of AhR suppresses pro-inflammatory pathways like NF-κB, reducing the release of pro-inflammatory cytokines [e.g., tumor necrosis factor α (TNF-α), IL-6]. Since intestinal inflammation is a potent inducer of ER stress, suppressing inflammation via AhR can indirectly alleviate inflammation-driven ER stress. Furthermore, it has been reported that AhR activation modulates ER stress through the downregulation of microtubule-associated protein light chain 3(LC3)II expression (38).

Through 16sRNA sequencing, *Muribaculaceae* was the most abundant bacterial family across all groups. Although *A. muciniphila* supplementation did not result in a statistically significant increase in its abundance, an upward trend was observed. Previous studies have indicated that *Muribaculaceae* is also closely associated with DCA (39) and its expansion has been shown to inhibit the p-IRE/XBP1 pathway (40). We identified that among the differentially abundant bacteria across the three groups (control, model, and AKK-treated), those with higher relative abundance were *Parasutterella and Desulfovibrio*. Numerous studies have reported an elevated abundance of *Desulfovibrio* in the gut of patients with UC, largely attributed to its production of cytotoxic hydrogen sulfide—a key factor in disrupting the intestinal mucus barrier and impairing epithelial cell function. The contradictory findings regarding *Desulfovibrio* in our study may be attributed to its environmental sensitivity. As shown by Xie et al. (41), although *D. vulgaris* exacerbates colitis, its colonization and abundance are modulated by specific inflammatory conditions. This environmental niche dependency suggests that while *Desulfovibrio* levels generally increase in UC, its survival can be suppressed under extreme inflammatory stress. Consequently, fluctuations in its abundance are not unidirectional but are dictated by inflammation severity. In the DSS-induced acute severe colitis model employed here, intense intestinal inflammation—along with associated microenvironmental alterations such as reactive oxygen species (ROS) bursts, pH shifts, and antimicrobial peptide secretion—likely contributed to the transient suppression of certain strict anaerobes, including *Desulfovibrio*. We found that the application of *A. muciniphila* could reduce the Firmicutes/Bacteroidetes (F/B) ratio in the model group and adjust the intestinal flora homeostasis, in which *Parasutterella* was significantly reduced after administration. Studies have shown that this genus is closely related to bile acid metabolism. The intervention of *Parasutterella* can significantly reduce the content of DCA and LCA, inhibit the expression of FXR and its downstream target genes small heterodimer partner (SHP) and fibroblast growth factor 15 (FGF15), and reduce the levels of bile acid transporters organic solute transporter β (OSTβ) and ileal bile acid-binding protein (IBABP) (33). Since the application of *A. muciniphila* significantly reduced the abundance of *Parasutterella*, we further examined the changes in fecal BAs in mice before and after *A. muciniphila* intervention, and the results showed that the levels of DCA, UDCA, TCA, and TCDCA were decreased in the model group, consistent with previous reports (42). Following *A. muciniphila* supplementation, the concentrations of these bile acids were elevated. Among these four bile acids, DCA has been reported to activate ATF4 in the context of chronic kidney disease (43), thereby promoting ER stress. The involvement of the other two major ER stress signaling pathways, however, remains less explored. UDCA inhibits ER stress (44–49) and has been shown to significantly increase the abundance of *A. muciniphila* (45). TCA upregulates ER stress, which subsequently triggers apoptosis in epithelial cells (50, 51). While no direct evidence links TCDCA to ER stress modulation, its enzymatically derived product, TUDCA, was significantly increased upon *A. muciniphila* supplementation in the DSS model. Previous studies have shown that DCA, LCA, and TCA in the gut can activate FXR, and TUDCA is a clinically available ER stress inhibitor (52). These findings suggest that *A. muciniphila* modulates the levels of various BAs in DSS-induced colitis, which can either promote or inhibit ER stress. Therefore, the net effect of *A. muciniphila* on ER stress in experimental colitis requires elucidation. Furthermore, FXR, a pivotal target of bile acid signaling, plays a key role in maintaining ER homeostasis, and this FXR-mediated mechanism may contribute to the anti-colitis action of *A. muciniphila*.

We detected the related indexes of ER stress in the colon tissues of each group, and the results showed that ER stress markers including p-IRE1α and XBP1 were markedly enhanced in colons after 10 days of DSS treatments, which aligns with previous findings (16, 53). Administration of *A. muciniphila* significantly downregulated p-IRE and XBP1u expression levels compared to the DSS-induced colitis group. Recently, the role of the IRE1α signaling pathway in UC has received more attention (54). Notably, experimental evidence demonstrates dynamic regulation of this pathway during disease progression, with acute-phase upregulation observed following short-term DSS exposure and subsequent downregulation during chronic disease stages after prolonged DSS treatment (55). IRE1 is a transmembrane protein with protein kinase and endoribonuclease activities. Unfolded proteins in the ER stimulate IRE1 phosphorylation, which activates endoribonuclease activity. Upon activation, IRE1 splices the substrate precursor XBP1 mRNA intron, producing an active XBP1 protein. p-IRE1 can also catalyze the signaling pathway mediated by c-Jun N-terminal protein kinase (JNK) and caspase-12 through TNF receptor-associated factor 2 (TRAF2)to promote apoptosis (56, 57), which are essential for maintaining ER function. XBP1 protein controls the transcription of a group of UPR target genes, including protein disulfide isomerase (PDI) (58), and regulates the expression of CHOP, which is involved in the occurrence and development of UC by inducing apoptosis through a mitochondria-dependent pathway. XBP1 is a classic ER stress signaling molecule, existing in two forms: XBP1s and XBP1u. The unconventional splicing of XBP1 mRNA is orchestrated at multiple levels and is linked to the transient expression of the unspliced form, XBP1u. Despite its high instability and rapid degradation by the 26S proteasome during translation, XBP1u utilizes a well-conserved hydrophobic domain to obstruct ribosomes on the ER membrane, thereby promoting the cytosolic splicing of its own mRNA and enabling its delivery to the IRE1 splicing site (59). Murine cytomegalovirus (MCMV) activates the IRE1–XBP1 pathway early after infection to relieve the suppression by XBP1u, the product of unspliced mRNA. XBP1u inhibits viral gene expression and replication by preventing XBP1s and ATF6 from activating the viral major immediate-early promoter (60). To our knowledge, no studies have directly investigated the role of XBP1u in the development or therapy of colitis. Combined with the above results, it indicates that *A. muciniphila* can alleviate the symptoms of DSS-induced acute colitis in mice by targeting both IRE1α and XBP1u within the ER stress signaling pathway.

Previous studies have shown that the activation of FXR can suppress intestinal inflammation, improve colon shortening, and weight loss in colitis mice. In this study, the intervention of *A. muciniphila* changed the size and composition of the bile acid pool in experimental colitis, impacting the response to pharmacologic FXR activation. To verify the critical role of FXR in *A. muciniphila* treatment of UC and its regulation of ER stress, we conducted DSS-induced colitis based on FXR$^{-/-}$ mice and used *A. muciniphila* for intervention treatment. The results showed that after the deletion of FXR, the therapeutic effect of *A. muciniphila* was significantly weakened, the symptoms such as weight loss, blood in the stool, and diarrhea in mice could not be improved, and the colon length and pathological conditions were not alleviated. In terms of pro-inflammatory factor expression, in FXR-deficient colitis mice, the inhibitory effect of *A. muciniphila* on IL-1β and IL-6 expression in colon tissue and serum was weakened. In FXR$^{-/-}$ colitis mice, the effects of *A. muciniphila* on enhancing the mucus layer and increasing the expression of MUC2 were largely suppressed. Current evidence regarding the mechanistic linkage between the FXR pathway and IRE1α/XBP1 signaling remains contradictory. Pharmacological activation of FXR has been shown to upregulate XBP1 splicing efficiency and p-IRE1α levels in murine (25), while the comparative analysis revealed significant activation of hepatic XBP1 and associated UPR effectors in aged FXR$^{-/-}$ mice (24-week-old vs 10-week-old), a phenomenon absents in WT counterparts (61). In the present study, the only FXR-dependent change among ER stress-related markers was observed in the expression of XBP1u, which was significantly elevated in the DSS model group and reduced after *A. muciniphila* administration. Quantitative

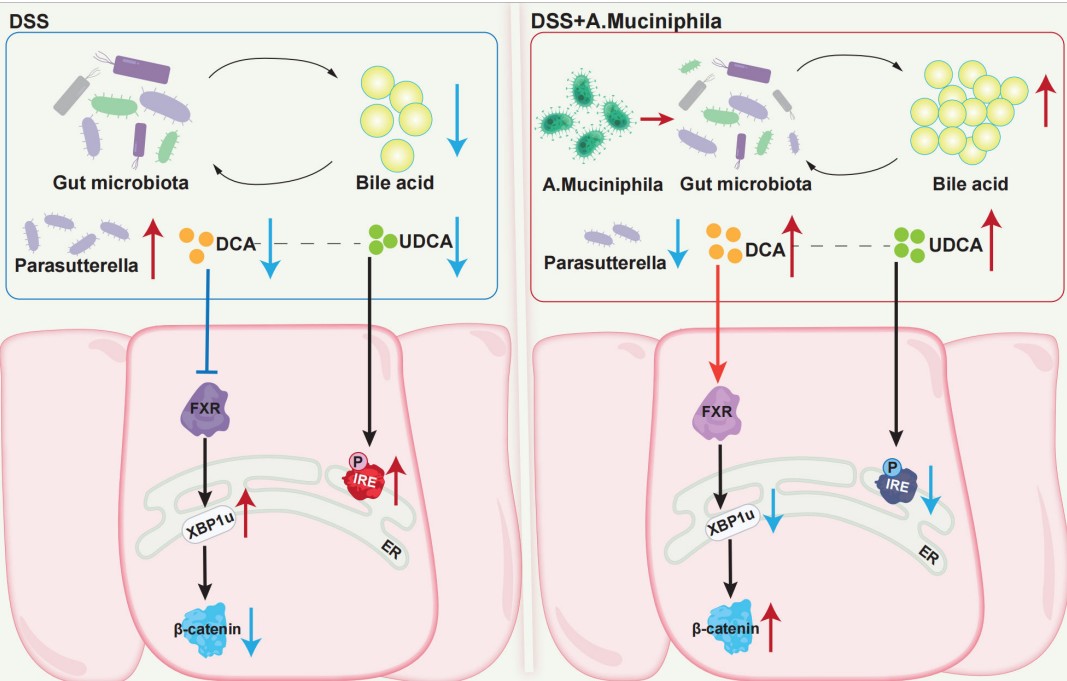

**FIG 7** *A. muciniphila* ameliorates DSS-induced colitis through a mechanism involving gut microbiota remodeling, modulated bile acid metabolism, and the alleviation of colonic epithelial ER stress, via both FXR-mediated regulation of XBP1u and FXR-independent suppression of p-IRE1α.

analysis revealed a marginally significant upward trend in XBP1u expression (1.85-fold, $P = 0.06$) in our FXR$^{-/-}$ mice compared with wild-type controls, as evidenced by protein immunoblotting. Although the role of XBP1u in colitis has not been directly reported, a previous study (35) indicated that it promotes ubiquitination-mediated degradation of β-catenin. β-Catenin is a key component of the intestinal barrier under physiological conditions. Consistent with this, β-catenin expression was downregulated in the DSS model and restored following intervention (62). In addition, the FXR agonist GW4064 has been shown to enhance β-catenin protein levels, promote osteoblast differentiation, and activate the Wnt/β-catenin signaling pathway in endothelial cells, contributing to the repair of intestinal vascular mucosal barrier integrity, reduction of bacterial translocation to the liver, and attenuation of hepatic inflammation. Our measurements of colonic β-catenin expression revealed a significant decrease in DSS-induced colitis mice, which was markedly rescued by *A. muciniphila* supplementation in an FXR-dependent manner. In contrast, the inhibition of p-IRE was not FXR-dependent. Although FXR agonists like DCA decreased after DSS induction, p-IRE1 expression was unexpectedly upregulated, further supported by the fact that DSS did not significantly elevate p-IRE1 in FXR-deficient mice. This observation may be explained by the involvement of another key bile acid, UDCA. Unlike DCA, UDCA does not activate FXR (18) yet significantly inhibits ER stress. In DSS-induced mice, UDCA was markedly downregulated, a change which was reversed by *A. muciniphila* supplementation. Previous studies have shown that BAs can alleviate ER stress via alternative receptors, particularly the Takeda G protein-coupled receptor 5 (TGR5) pathway (63). This offers a plausible mechanism for the FXR-independent inhibition of p-IRE1 observed in our study.

This study elucidates that *A. muciniphila* ameliorates DSS-induced murine colitis through FXR-mediated regulation of bile acid metabolism and subsequent suppression of the XBP1u ER stress pathway. While the investigation establishes FXR's central role in mediating this process, several potential mechanisms and research directions warrant further exploration. Notably, *A. muciniphila* significantly reduces the abundance of *Parasutterella*, a genus previously reported to inhibit secondary bile acid biosynthesis (e.g., DCA, LCA). The mechanistic basis of this microbial interaction requires clarification

—specifically, whether *A. muciniphila* directly suppresses *Parasutterella* or indirectly modulates its abundance through bile acid profile alterations. The safety of orally administered *A. muciniphila* in humans has been evaluated in clinical trials, primarily for metabolic conditions. The study by Depommier et al. demonstrated that daily oral supplementation with up to $10^{10}$ CFU of *A. muciniphila* (both live and pasteurized forms) for 3 months was safe and well-tolerated in overweight/obese insulin-resistant volunteers, with no serious adverse events reported (64). The bacterium is typically administered in capsules or suspended in a carrier medium to ensure viability and delivery through the gastrointestinal tract to its site of action in the colon. While clinical results are promising, further phase II/III trials specifically in UC patients are warranted to formally establish its efficacy, optimal dosing, and long-term safety for this indication. Although murine experiments demonstrated dose-dependent therapeutic efficacy (with superior outcomes at $3 \times 10^8$ CFU), it will be worth investigating the proper concentration range of *A. muciniphila* that could modulate ER homeostasis in normal colon physiology and during colitis. Given the dual tissue-specific expression patterns of FXR in hepatic and intestinal systems, combined with the established pathogenic significance of enterohepatic bile acid cycling in colitis progression, we strategically employed FXR knockout (FXR-KO) models for this investigation. It should be noted, however, that hepatic-specific FXR ablation might introduce partial confounding variables requiring careful interpretation of experimental outcomes. While this work provides compelling evidence for the FXR-endoplasmic reticulum stress axis in *A. muciniphila*-mediated colitis amelioration, translational validation through multidimensional approaches remains essential to assess its clinical potential.

In conclusion, *A. muciniphila* supplementation significantly altered the gut microbiota, most notably by inducing a bloom of *Parasutterella*—which demonstrated a negative correlation with fecal DCA levels. Administration of *A. muciniphila* reversed the colitis-induced alterations in the size and composition of the bile acid pool, primarily by counteracting the reduction in DCA and UDCA levels. The anti-colitis effect of *A. muciniphila* was exerted by maintaining ER stress homeostasis, via both FXR-mediated regulation of XBP1u and FXR-independent suppression of p-IRE1α (Fig. 7). Our data suggested that *A. muciniphila* supplementation in appropriate dose might extend the therapeutic benefit for the treatment of UC.

## ACKNOWLEDGMENTS

This study was supported by the National Natural Science Foundation of China (Grant No. 82305241). This study was also supported by Zhejiang Provincial Traditional Chinese Medicine Science and Technology Plan (2024ZL005). This study was also supported by a Project Funded by the Priority Academic Program Development of Jiangsu Higher Education Institutions (PAPD) and Jiangsu Provincial Medical Key Laboratory (ZDXYS202208).

The investigation and methodology were completed by Q.W. and Y.C. F.B., K.Z., B.S., and L.H. performed the experiment, wrote an original draft, and revised the manuscript. Z.L., X.Y., and C.C. assessed the data for potential analysis. F.J., Y.T., W.Z., and D.Z. contributed reagents/materials/analytical tools and reviewed the draft.

## AUTHOR AFFILIATIONS

[1]Affiliated Hospital of Nanjing University of Chinese Medicine, Nanjing, Jiangsu, People's Republic of China

[2]Jiangsu Province Hospital of Chinese Medicine, Nanjing, Jiangsu, People's Republic of China

[3]Department of Colorectal Surgery, Zhejiang Provincial People's Hospital, Affiliated People's Hospital of Hangzhou Medical College, Hangzhou, China

[4]Department of Colorectal Surgery, The Affiliated Hospital of Nanjing University of Chinese Medicine, Nanjing, China

[5]No.1 Clinical Medical College, Nanjing University of Chinese Medicine, Nanjing, People's Republic of China

[6]Department of Basic Pharmacology, The Affiliated Hospital of Nanjing University of Chinese Medicine, Nanjing, China

## AUTHOR ORCIDs

Fan Bu  http://orcid.org/0009-0009-7945-0576
Yugen Chen  http://orcid.org/0009-0005-2510-4054
Qiong Wang  http://orcid.org/0000-0001-6251-4996

## FUNDING

| Funder | Grant(s) | Author(s) |
| --- | --- | --- |
| National Natural Science Foundation of China | 82305241 | Fan Bu |
| ZHEJIANG PROVINCIAL TRADITIONAL CHINESE SCIENCE AND TEHNOLOGY PLAN | 2024ZL005 | Fan Bu |
| A PROJECT FUNDED BY THE PRIORITY ACADEMIC PROGRAM DEVELOPMENT OF JIANGSU HIGHER EDUCA-TION INSTITUTIONS | PAPD | Qiong Wang |
| JIANGSU PROVINCIAL MEDICAL KEY LABORATORY | ZDXYS202208 | Yugen Chen |

## AUTHOR CONTRIBUTIONS

Fan Bu, Funding acquisition, Writing – original draft | Kaiqing Zhang, Software | Bingbing Song, Software | Linhai He, Software | Zhihua Lu, Data curation | Xiaomin Yuan, Data curation | Chen Chen, Software | Feng Jiang, Formal analysis | Yu Tao, Formal analysis | Wei Zhang, Methodology | Dan Zhang, Investigation | Yugen Chen, Methodology | Qiong Wang, Conceptualization, Methodology, Writing – review and editing

## DATA AVAILABILITY

The fecal metabolomics data have been deposited in MetaboLights the under accession number MTBLS13104 (https://www.ebi.ac.uk/metabolights/MTBLS13104). Other supporting data are available within the article or from the corresponding authors upon reasonable request.

## ADDITIONAL FILES

The following material is available online.

### Supplemental Material

**Fig. S1 (mSystems01589-25-s0001.docx).** Expression levels of ATF6.

### Open Peer Review

**PEER REVIEW HISTORY (review-history.pdf).** An accounting of the reviewer comments and feedback.

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
