## [Reviewer comments · mSystems]

***Akkermansia muciniphila* alleviates experimental colitis through FXR-mediated repression of Unspliced XBP1**

fan bu, Kaiqing Zhang, Bingbing Song, Linhai He, zihua Lu, chen chen, xiaomin Yuan, feng Jiang, yu Tao, Wei Zhang, dan Zhang, yugen Chen, and qiong wang

Corresponding Author(s): fan bu, Affiliated Hospital of Nanjing University of Chinese Medicine

Review Timeline:

Submission Date:

November 24, 2025

Accepted:

December 5, 2025

Editor: Batbileg Bor

Reviewer(s): Disclosure of reviewer identity is with reference to reviewer comments included in decision letter(s). The following individuals involved in review of your submission have agreed to reveal their identity: yi yin (Reviewer #2)

Transaction Report:

DOI: <https://doi.org/10.1128/msystems.01589-25>

Re: mSystems01589-25 (***Akkermansia muciniphila* alleviates experimental colitis through FXR-mediated repression of Unspliced XBP1**)

Dear Dr. fan bu:

Your manuscript has been accepted, and I am forwarding it to the ASM production staff for publication. Your paper will first be checked to make sure all elements meet the technical requirements. ASM staff will contact you if anything needs to be revised before copyediting and production can begin. Otherwise, you will be notified when your proofs are ready to be viewed.

Sincerely,
Batbileg Bor
Editor
mSystems

Reviewer #2 (Comments for the Author):

The authors have addressed all my comments.